# Comparison of Static and Dynamic Assays When Quantifying Thermal Plasticity of Drosophilids

**DOI:** 10.3390/insects11080537

**Published:** 2020-08-15

**Authors:** Christian Winther Bak, Simon Bahrndorff, Natasja Krog Noer, Lisa Bjerregaard Jørgensen, Johannes Overgaard, Torsten Nygaard Kristensen

**Affiliations:** 1Section of Biology and Environmental Science, Department of Chemistry and Bioscience, Aalborg University, Fredrik Bajers Vej 7H, DK 9220 Aalborg, Denmark; sba@bio.aau.dk (S.B.); nkn@bio.aau.dk (N.K.N.); tnk@bio.aau.dk (T.N.K.); 2Zoophysiology, Department of Bioscience, Aarhus University, C.F. Møllers Alle Building 1131, DK 8000 Aarhus C, Denmark; lbj@bios.au.dk (L.B.J.); johannes.overgaard@bios.au.dk (J.O.)

**Keywords:** dynamic and static thermal assays, thermal tolerance landscapes, heat tolerance, hardening, CTmax

## Abstract

**Simple Summary:**

Temperature directly affects many biological processes, from enzymatic reactions to population growth, and thermal stress tolerance is central to our understanding of the global distribution and abundance of species and populations. Given the importance of thermal stress tolerance in ecophysiology and evolutionary biology it is important to be able to measure thermal stress resistance accurately and in ecologically relevant ways. Several methods for such quantification exist in the arthropod literature and the comparability of different methods is currently being debated. Here we reconcile the two most commonly used thermal assays (dynamic ramping and static knockdown assays) for quantifying insect heat tolerance limits and plastic responses using a newly suggested modeling technique. We find that results obtained on the basis of the two assays are highly correlated and that data from one assay can therefore reasonably well predict estimates from the other. These data are of general relevance to the study of thermal biology of ectotherms.

**Abstract:**

Numerous assays are used to quantify thermal tolerance of arthropods including dynamic ramping and static knockdown assays. The dynamic assay measures a critical temperature while the animal is gradually heated, whereas the static assay measures the time to knockdown at a constant temperature. Previous studies indicate that heat tolerance measured by both assays can be reconciled using the time × temperature interaction from “thermal tolerance landscapes” (TTLs) in unhardened animals. To investigate if this relationship remains true within hardened animals, we use a static assay to assess the effect of heat hardening treatments on heat tolerance in 10 *Drosophila* species. Using this TTL approach and data from the static heat knockdown experiments, we model the expected change in dynamic heat knockdown temperature (CT_max_: temperature at which flies enter coma) and compare these predictions to empirical measurements of CT_max_. We find that heat tolerance and hardening capacity are highly species specific and that the two assays report similar and consistent responses to heat hardening. Tested assays are therefore likely to measure the same underlying physiological trait and provide directly comparable estimates of heat tolerance. Regardless of this compliance, we discuss why and when static or dynamic assays may be more appropriate to investigate ectotherm heat tolerance.

## 1. Introduction

Quantification of arthropod thermal tolerance requires assays that are sensitive to the treatment effects of interest whilst also exposing the animals to conditions of ecological relevance. Several such assays exist and a discussion on how choice of experimental conditions can affect assay outcome is ongoing [1,2,3,4,5,6]. The effect of temperature on physiological performances is typically described by thermal tolerance curves with upper and lower critical endpoints designating respective high and low temperatures where physiologically performance reaches zero. Such tolerance curves are often generated using one of two types of thermal exposure protocols within arthropods: the static knockdown assay [7] and the dynamic ramping assay [8]. Static knockdown assays expose individuals to an abrupt, constant and stressful temperature and examine the time taken to reach a predetermined “failure” endpoint. The rate of “injury accumulation” in the static assay is assumed to be constant but temperature dependent, and heat knockdown time is recorded when the critical amount of “injury” has accumulated (Figure 1A). In the dynamic ramping assay, animals are initially exposed to a benign temperature followed by a constant increase (or decrease) in temperature. In the dynamic assay, the rate of “injury” accumulation will increase exponentially until a critical amount of “injury” has accumulated and the temperature of knockdown (CT_max_) is recorded (Figure 1B). The exponential nature of this time × temperature relationship is well described in the literature and it has been suggested that the exponent, which can be deduced from TTLs, describes the relation between knockdown time and temperature (i.e., the intensity of thermal stress) [1,9]. The TTL theory suggests an underlying relationship between the temperature intensity, exposure time and “injury” accumulation [10] and predicts improved critical thermal limits with increasing temperature ramping rates. A recent publication noted that this prediction is not universally observed within arthropods and accordingly the usage of TTL is still undergoing debate [6].

Irrespective of the method chosen, determining a universally true thermal tolerance is a complex endeavor as such estimates will vary with numerous factors such as photoperiod [11,12], ontogeny [13,14], time of the day [15,16,17], rearing temperature [18,19,20,21], hardening [22,23,24], animal density [25,26] and applied anesthetics [27,28]. Elements within the assay itself are also important and include the chosen exposure temperature within the static assay [29]; the initial temperature and rate of temperature change within the dynamic assay [30,31,32,33,34]; the diversity of thermal endpoint measures to choose from (often defined behaviorally as coma, loss of coordinated movements, onset of muscle spasms, death or a measure of recovery from coma) [8,10,35]. 

It has been debated if the static and dynamic assays impose the same type of stress (i.e., quantify the same underlying thermal trait) and if it is valid to compare results obtained across studies and assay types [1,2,3,9,36,37]. The principle argument in favor of dynamic assays is that they better reflect the natural temperature fluctuations that individuals might encounter within their natural environment and therefore better consolidate activity capacity ranges that organisms can endure in the field [3]. Conversely, it has been suggested that the slower nature of the dynamic assays could introduce confounding factors such as desiccation or starvation during heat stress assays [1]. However, empirical studies do not support this claim [3,36,38]. Furthermore, it has been argued that hardening responses might be activated during assays with slow rates of temperature change since more time is allowed for physiological adjustments to occur [7,39,40]. Historical studies on fishes [41] and a recent study on insects [9] have argued that the two assays (static and dynamic) can be reconciled mathematically if the exponential relationship between knockdown time and temperature is known. 

A strong link between dynamic and static assays has previously been demonstrated using *Drosophila* species taken directly from their normal rearing temperature (non-hardened animals) [9]. Here, we further investigate this link, by exposing 10 *Drosophila* species to two heat hardening treatments and a standard rearing temperature and test if assays still convey comparable and predictable estimates across treatments. For all species, we obtained empirical measurements of heat tolerance with or without prior heat hardening using both static and dynamic assays. These data allow for a simple comparison of treatment and species’ effects when measured using the two assays. In addition, we used the data from static assays (heat knockdown time, HKDT) together with a literature-based exponent of the survival time × temperature relationship to model a “predicted CT_max_” in a dynamic assay. By comparing the hardening response of this predicted CT_max_ with the observed hardening responses in empirically measured CT_max_, we could evaluate if the two methods provide directly comparable estimates of heat tolerance.

## 2. Methods and Materials:

### 2.1. Animal Husbandry

Ten *Drosophila* species (see Table 1 for details), randomly chosen across the Drosophila phylogeny [42], were obtained and kept under common garden conditions for at least 3 generations prior to experimentation: roughly 200 adult flies were placed in 250 mL bottles containing 35 mL standard yeast-sucrose-agar fly medium [43] and kept in a thermo-cabinet at 20 °C and a 12:12 day:night cycle. Bottles were density controlled with parental flies producing eggs for a species-specific time period (ranging from 12 to 48 h) giving similar egg, larvae and adult fly densities across bottles and species (150–250 adults emerging per bottle). Prior to each experimental run, 30 newly emerged male flies of each species were (within a 12-h window post eclosion) sampled under mild CO_2_ induced anesthesia (<5 min) and transferred to 20 mL vials containing 7 mL standard fly medium where they recovered for five consecutive days. Accordingly, flies used for this experiment were roughly six-day-old adult males at the time of experimentation.

### 2.2. Experimental Protocol

Flies were tested within one week in six replicate runs for each assay (CT_max_ and HKDT), with each species and treatment group combination present in each run (*D. montana* was excluded from the first two runs as an insufficient number of flies had developed). Flies were evenly split into three groups prior to heat tolerance tests; non-hardened controls and two groups hardened at 31 or 33 °C, respectively. Flies from each species and from all treatment groups were transferred to screw-capped glass vials and exposed to their respective temperatures (20 °C in their incubator and 31 or 33 °C in respective water baths) for 1 h followed by 70 min of recovery at 20 °C. Following recovery, vials were randomly fixed onto a holding-rack and submerged into a see-through water bath set to either a static temperature of 38 °C (HKDT) or a ramping temperature starting at 20 °C with an incremental increase of 0.1 °C min^−1^ (CT_max_). For each treatment, we tested ∼15 flies of each species over the six experimental runs (for exact numbers see Appendix A). We chose neuromuscular coma as the thermal endpoint where, when neither optical nor mechanical encouragement (flashes of light or taps from a metal rod, respectively) could elicit visually observable twitches, the time (HKDT) or temperature (CT_max_) was recorded as the tolerance threshold for that individual. Each individual fly would be systematically inspected roughly once every 10 s whilst submerged within the bath.

### 2.3. Data Analysis

All data analyses were performed in *R* version 3.6.2 [44], with the critical value for significance set to 0.05. Recordings of HKDT and CT_max_ were screened for outliers by calculating the median absolute deviation (MAD) within each treatment × species combination using the function mad() from the Stats package in base *R*. Values that differed more than ±3 MAD from the median were discarded [45], resulting in the removal of 22 HKDT values (out of 556) and 21 CT_max_ values (out of 546) evenly distributed among the species × treatment combinations. A linear regression (lm()-function) was used to test for correlation between dynamic (CT_max_) and static (log_10_ HKDT) heat tolerance measures. The linear regression was evaluated using summary(), and the F-statistics on the regression coefficient was used to test if the slope was different from zero [46]. Finally, to test if hardening altered heat tolerance within each species, we performed a one-way ANOVA to examine if CT_max_ or HKDT differed between the three treatments: control, 31 °C hardened or 33 °C hardened animals. If a significant effect of treatment was found, a multiple comparison test between hardening treatments and the control was performed using a Dunnett post hoc test (glht() and linfct = mcp(Treatment = “Dunnett”)) in the multcomp-package [47].

### 2.4. Predicting CT_max_ from HKDT (Heat Knockdown Time)

A thermal tolerance landscape (TTL) is argued to carry information on a species’ relative rate of heat injury accumulation at different temperatures, and this exponential factor is represented in the parameter *z* (*z* is the negative reciprocal of the TTL slope, see [10] and [9] for a discussion of TTL). For mathematical modeling of data, we used the mean value of *z* = 2.534 previously recorded for five *Drosophila* species that are also present in the current study (*D. equinoxialis*, *D. immigrans*, *D. mercatorum*, *D. montana* and *D. rufa*) [9]. From this, we can mathematically model the exponential function describing heat injury accumulation rate at different temperatures for each species (see Equation (7) in [9]). With the known HKDT of a species we can therefore also predict the CT_max_ of that species exposed to a given treatment prior to testing (20, 31 or 33 °C), assuming that both static and dynamic assays require the same amount of critical injury before CT_max_ is reached (graphically presented in Figure 1 as similar “areas” of accumulated injury in static and dynamic assays). 

Using this approach, we made predictions of CT_max_ based on mean HKDT from each treatment (control, 31 °C, 33 °C) within each species in the static assays using the following equation:(1)$$CT_{max}=\frac{1}{k}\ln\left[kb\cdot HKDT\cdot e^{kT}+e^{kT_{C}}\right]$$
where *k* is $\frac{\ln\left(10\right)}{z}$ (*z*, and accordingly *k*, is unitless), *b* is the ramp rate (0.1 °C min^−1^), *HKDT* is the measured knockdown time in the static assay, *e* is Euler’s number, *T* is the temperature of the static assay (38 °C) and *T_c_* is an estimated “critical” temperature, above which heat injury rate is assumed to surpass repair rate, resulting in accumulation of heat injury. *T_c_* is calculated for each species using the common value of *z* and the mean HKDT of non-hardened controls to extrapolate to the temperature that would result in knockdown after 24 h exposure. 

With this function, we calculated a predicted CT_max_ for each species × treatment combination. We then calculated the predicted change in CT_max_ between the non-hardened control and the hardened animals. This difference (∆CT_max-predicted_) based on data from the static assay could then be compared to the difference empirically observed from the dynamic CT_max_ trials of hardened and control flies (∆CT_max-observed_). Similarity of ∆CT_max-predicted_ and ∆CT_max-observed_ will therefore support the suggestion that static and dynamic assays are essentially measuring the same underlying physiological trait. To test this hypothesis, we performed a linear regression on ∆CT_max-observed_ against ∆CT_max-predicted_ and tested if this relationship was different from the line of unity (intercept = 0, slope = 1), using LinearHypothesis() from the *R* package Car [48]. 

## 3. Results

Both the static and the dynamic heat tolerance assays revealed considerable differences between species. HKDT in non-hardened controls varied more than four-fold between species (range: 254.5 to 1206.4 s) and CT_max_ varied more than 2.1 °C between species (range: 36.8 °C to 39.0 °C) (Table 2). The heat tolerance measurement from the dynamic CT_max_ and static HKDT assays were positively correlated and hardening treatments did not alter this positive and significant association (Figure 2A). Hardening treatments at either 31 or 33 °C resulted in both increased and decreased heat tolerance depending on species and treatment (Table 2). With the static assay the hardening response caused changes in HKDT ranging from −35% to +69% relative to control HKDT, and these changes were significant in 4 out of the 20 hardening treatments × species combinations (evaluated with one-way ANOVA, with a Dunnett post hoc test applied). In the dynamic ramp test, hardening resulted in observed absolute changes in CT_max_ ranging from −0.7 °C to +0.3 °C, but these changes did not reach the level of statistical significance.

Under the assumption that there is an exponential increase in the rate of injury accumulation with increasing temperature, we modelled data from the static assay (HKDT) to make predictions of CT_max_ (Table 2). The predicted CT_max_ based on HKDT values ranged from 37.0 °C to 38.7 °C in the control group and there was a significant and positive association between predicted CT_max_ and empirically measured CT_max_ (*R*^2^ = 0.59, *p* = 0.009) (Figure 2B). This approach was also used to predict how hardening at either 31 or 33 °C would change CT_max_ (Pred. ∆CT_max_). Results showed that there is a significant and positive association between Pred. ∆CT_max_ and Obs. ∆CT_max_ (Figure 2C). The *R*^2^ value of this linear regression is 0.43 and the slope is significantly different from both 0 (*p* < 0.01) and 1 (*p =* 0.026) while the intercept is not different from 0 *(p =* 0.106). 

## 4. Discussion

Measures of heat knockdown time and critical thermal maxima provide tangible means for evaluating organismal thermal tolerance. Such estimates are widely utilized in attempts to predict the impact of climate change on species distribution, evolutionary and physiological responses to thermal stress and extinction events [49,50]. Here we investigated heat tolerance and heat hardening capacities within a range of *Drosophila* species employing both static and dynamic thermal assays. 

Our study reveals large variation in heat tolerance and hardening responses among the tested *Drosophila* species (Table 2). Species under investigation responded both adaptively and maladaptively to hardening (increasing or decreasing HKDT or CT_max_ estimates compared to their control group, respectively). This illustrates several important methodological aspects when assessing thermal tolerance. Firstly, we used a standardized hardening treatment across species and therefore ignored that the basal thermal tolerance is species specific. Nyamukondiwa et al. [51] showed that acute heat tolerance within several *Drosophila* species was improved only when species were pretreated at temperatures close to their respective upper thermal limits. Thus, the hardening treatments in this study were likely insufficiently stressful to elicit a physiological response in the most heat tolerant species investigated here. In addition, heat sensitive species are likely to sustain and accumulate injury during the hardening treatments that we used, which could lead to an unintended lower measurement of heat tolerances in these “hardened” flies. This study therefore provides only measures of hardening potential under the hardening conditions experienced, which are unlikely to reflect hardening potential of the different species. We argue that this is a complication that is often ignored in experimental studies of hardening and acclimation responses in ectotherms [31].

Comparative studies of thermal tolerance performed on arthropods have employed various methods to assess heat or cold resistance [3,9,30,52,53]. Here, we found that results from static and dynamic assays for heat tolerance were highly correlated (Figure 2A) which suggest that both assays are indicative of the same underlying thermal trait. Our results support that inherent measures of heat tolerance are correlated across assays and that ranking of species are robust across different assay conditions. Although the two assays measure different parameters (a time (HKDT) or a temperature (CT_max_)) our modeled data show that the data output from one assay (HKDT) can provide a reasonable estimate of the other assay (CT_max_). This conclusion is based on the positive correlation between the observed and predicted CT_max_ (*R*^2^ = 0.59) (Figure 2B). In our calculations (see Equation (1)) we assumed a similar exponential relationship between temperature and the rate of injury accumulation for all 10 species (the same value of *z*). However, species differ in their thermal sensitivity of injury accumulation (*z*) and it is therefore likely that our predictions of CT_max_ would be even closer to the observed if we could use species-specific values of *z* (this would require *z* to be estimated by performing measurements of HKDT at a range of stressful temperatures). For example, Jørgensen et al. [9] found that predicted and observed CT_max_ were indistinguishable across a range of ramping rates when using *Drosophila* species-specific values of *z*. 

In the present study, we focused our comparison of static and dynamic assays by comparing the predicted (based on static data) and observed change in CT_max_ following the hardening treatments. It is therefore only the added effect of hardening that is sensitive to the selected value of *z* and our hypothesis was tested by investigating the relation between predicted ΔCT_max_ and observed ΔCT_max_ (Figure 2C). Our regression of observed versus predicted ΔCT_max_ data had a slope of 0.6, which was significantly different from the hypothesized value of 1. Based on this result, we cannot exclude the possibility that the two assays are measuring different physiological processes, but we did find the correlation to be positive and considering that the hardening responses are generally small we conclude that the dynamic and static assay are likely to report the same physiological dysfunction during heat stress. This conclusion is also supported by the general agreement of predicted and observed values of CT_max_ (Figure 2B). The findings of the present study are therefore in opposition to previous suggestions that ramping CT_max_ may be confounded by other physiological disturbances than heat stress alone (i.e., starvation or desiccation stress) [1]. It has also been proposed that dynamic assays are somewhat confounded by a “harden as you go” phenomenon, where the animals are allowed time to harden when slowly ramped up (or down) in temperature [22,39,40,54]. There is significant evidence that this occurs for some arthropods exposed to cold stress [22] and we cannot exclude that such hardening responses did also affect the relation between predicted ΔCT_max_ and observed ΔCT_max_ in the present study. We did not find the relationship between predicted ΔCT_max_ and observed ΔCT_max_ to follow the line of unity and found only significant differences between controls and hardened groups with the HKDT assay. Accordingly, results from the CT_max_ assay might be confounded by the “harden as you go” phenomenon, but, considering the general alignment between modeled and empiric CT_max_ values, we argue that this effect is relatively minor.

Despite the similarity of the conclusions reached from the two assay types investigated in the present study, there are still some practical differences between the assays that are important to acknowledge. In our experiments, the static assays typically lasted <30 min while the dynamic assays lasted >3 h. Comparably, the difference in time between the observation of HKDT or CT_max_ in the controls versus the hardened flies was typically <3 min. In dynamic assays, these are minute time differences considering that the entire measurement lasted ∼200 min. We therefore argue that it is often beneficial to use a static assay to measure small differences in heat tolerance between treatment groups. Thus, static assays can be adjusted to give a constant intensity of heat stress that allows for an appropriate time separation of treatments groups, if there is a difference (Figure 1A). In contrast, it is very impractical to use static assays to separate treatment/species that differ markedly in heat tolerance. For example, some species of *Drosophila* may survive 38°C for days whereas others succumb in mere minutes to this treatment. Such large differences in heat tolerance are easier to investigate with the dynamic assay. Here, the rate of heat injury accumulation increases exponentially with time and the same ramping assay therefore can be used to study species/treatment groups that have considerable differences in heat tolerance. The trade-off with dynamic assays is that the exponential increase in heat injury accumulation only allows a very short time-window to separate treatment groups that are marginally different (Figure 1B). These differences in sensitivity of different assays are also indicated in Table 2, where the small effects of hardening were never found to be significant in dynamic assays (likely due to a small signal-to-noise relationship), while hardening effects were occasionally found significant in the static assay. 

## 5. Conclusions

We conclude that the static and dynamic assays provide biologically (and mathematically) comparable results when compared across species with large heat tolerance variation, but caution that this conclusion has only been investigated for heat stress in insects. Future studies should therefore investigate if this is also true for cold stress protocols. We emphasize that the dynamic and static assays discussed here are all concerning heat stress protocols that last minutes to hours and it is unknown if/how these assays are related to the heat stress that can take place over days–months which can occur in nature. Although both dynamic and static assays are shown to generally measure the same “physiological failure” during heat stress, we argue that both offer unique benefits and limitations. Ramping assays are superior to investigate large differences between treatments or species, while static assays can be modified to focus on smaller differences between more similar treatment groups (i.e., hardening effects or population differences).

## Figures and Tables

**Figure 1 insects-11-00537-f001:**
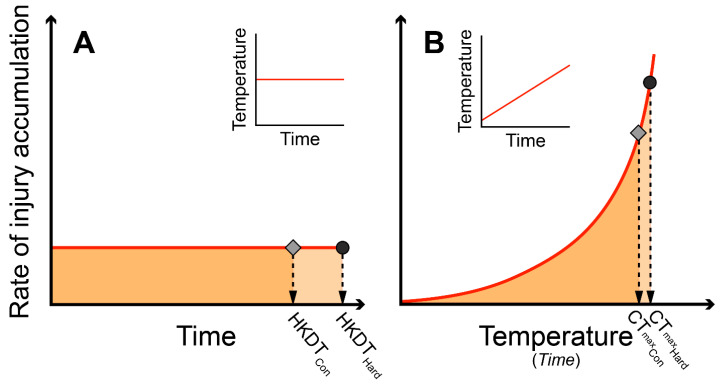
Theoretical development of heat injury in static and dynamic heat tolerance assays. (**A**) In a static assay, the animal is exposed acutely to a constant stressful temperature (see insert in (**A**)) and, hence, injury will accumulate at a constant rate until knockdown. The time of physiological failure in control animals is recorded as the heat knockdown time (HKDT_con_) and the accumulated injury during the assay is here depicted as the area below the curve. Heat hardening can change heat tolerance which can result in a higher HKDT (HKDT_Hard_). (**B**) In a dynamic heat tolerance assay the temperature is increased at a fixed rate (see insert in (**B**)). The rate of injury accumulation increases exponentially with temperature in a dynamic assay and heat tolerance is typically measured as the temperature of heat coma (CT_max_). Here, we use CT_max Con_ and CT_max Hard_ for control and hardened animals, respectively. If the two assays (static and dynamic) measure the same physiological response to heat stress, then it is possible to predict CT_max_ in a dynamic test from the HKDT in the static test. This analysis requires knowledge of the exponent that describes how the rate of injury accumulation increases exponentially with temperature, the rate of temperature change (ramp rate), and an assumption of the temperature above which injury starts to accumulate.

**Figure 2 insects-11-00537-f002:**
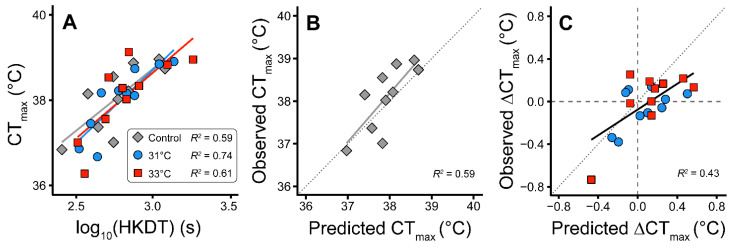
Comparison of static and dynamic heat tolerance measurements in 10 species of *Drosophila*. (**A**) Relationship between the logarithm of HKDT(s) from static tests and dynamic measures of CT_max_ (°C). Data for each species are recorded from control and two hardening treatment groups (Table 2). For each treatment, the coefficient of determination *R^2^* of linear regressions are displayed in the figure and all slopes were significantly different from 0 (*p* < 0.01). Treatment did not affect the slope and intercept of the linear regression between the two measures of heat tolerance (no significant interaction term or single effect of treatment). (**B**) CT_max_ in a dynamic assay was predicted mathematically for the 10 species using data from the static experiments. The positive and significant correlation between the predicted and measured CT_max_ (full line, *R*^2^: 0.59, *p* = 0.009) was not significantly different from the dotted line of unity (*p* = 0.559). (**C**) The predicted change in CT_max_ with hardening (predicted ∆CT_max_) was modeled from the hardening responses observed in the static assay and regressed against the observed change in CT_max_ following hardening (observed ∆CT_max_) (black line, see main text and Table 2). The *R^2^* value of this linear regression is 0.43 and the slope is significantly different from both 0 (*p* < 0.01) and 1 (*p =* 0.026) while the intercept is not different from 0 *(p =* 0.106). The dotted line represents the line of unity.

**Table 1 insects-11-00537-t001:** Origin and collection year of *Drosophila* species used in this study.

Species	Collection Location	Year Collected
*D. immigrans*	San Diego, USA	2016
*D. yakuba*	Liberia	1983
*D. equinoxialis*	Honduras	<1984
*D. sulfurigaster*	Finch Hatton, Australia	2013
*D. rufa*	Ehime, Japan	2016
*D. mercatorum*	Ehime, Japan	-
*D. simulans*	Ehime, Japan	2016
*D. birchii*	Mackay, Australia	2014
*D. lutescens*	Ehime, Japan	2016
*D. montana*	Finland	2008

Populations of the individual species are founded by merging a variable number of isofemale lines spanning from <10 to >200 for the different species. For further information on the species, see Appendix A.

**Table 2 insects-11-00537-t002:** Observed (Obs.) and predicted (Pred.) estimates of heat tolerance in 10 *Drosophila* species with and without prior heat hardening. Heat tolerance in static assays was measured as heat knockdown time (HKDT) at 38 °C and effects of heat hardening at 31 °C or 33 °C are reported as the % change in HKDT. Heat tolerance in the dynamic assay was measured as the temperature of heat knockdown (CT_max_) during a ramp test with a temperature increase of 0.1 °C min^−1^. The effects of heat hardening in dynamic tests are reported as the absolute change in CT_max_ from the control. Data from the dynamic assays are compared to predicted estimates of CT_max_ modelled from the static HKDT (see main text for further explanation). CT_max_ data predicted from the static measurements are shown in italics and statistically significant hardening effects are underlined and in bold.

	Static			Dynamic					
	Control	Hard. 31 °C	Hard. 33 °C	Control		Hard. 31 °C		Hard. 33 °C	
SPECIES	Obs. HKDT (s)	∆HKDT[%]		Obs. CT_max_ (°C)	Pred. CT_max_ (°C)	Obs. ∆CT_max_ (°C)	Pred. ∆CT_max_ (°C)	Obs. ∆CT_max_ (°C)	Pred. ∆CT_max_ (°C)
*D. equinoxialis*	552.8	−0.17	−0.07	38.55	37.83	−0.379	−0.195	−0.017	−0.076
*D. rufa*	694.1	0.10	0.17	38.21	38.08	−0.105	0.101	0.122	0.174
*D. immigrans*	254.5	0.30	0.27	36.84	36.98	0.021	0.281	0.168	0.257
*D. montana*	1206.4	−0.08	0.53	38.74	38.69	0.111	−0.094	0.217	0.460
*D. mercatorum*	749.6	0.02	−0.07	38.87	38.17	−0.134	0.023	0.253	−0.076
*D. yakuba*	554.6	−0.22	−0.35	37.01	37.83	−0.337	−0.261	−0.734	−0.469
*D.* *sulfurigaster*	376.9	0.59	0.69	38.15	37.40	0.073	0.503	0.133	0.566
*D.* *simulans*	1102.1	0.25	0.14	38.96	38.58	−0.058	0.244	−0.131	0.140
*D. birchii*	593.6	0.14	0.14	38.02	37.90	0.140	0.140	0.001	0.139
*D. lutescens*	440.3	−0.11	0.12	37.37	37.57	0.084	−0.122	0.189	0.120

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
