# Peer review of "Comparison of Static and Dynamic Assays When Quantifying Thermal Plasticity of Drosophilids"

_insects, 2020, doi:10.3390/insects11080537_

Round 1
Reviewer 1 Report
In this MS, Bak et al. performed comparison of heat tolerance and the effect of hardening in 10 Drosophila species to evaluate a more appropriate assay. The results are valuable for elucidating the mechanisms of heat tolerance in insects. I would recommend it for acceptance after the a few minor points listed below are addressed.
1. I know that researches using in Drosophila species (especially melanogaster) have greatly contributed to the understanding of thermal tolerance in insects. However, based on the experimental design and results of this MS, I think it is inappropriate to use that word of “arthropods” in the title. I recommend to change the title.
2. In Materials and methods parts (L 122-134), please explain the methods to measure time and temperature to enter heat come in more detail. For example, how often did they check the condition of the flies in the bottle? Did they take the bottle out of the water bath for observation?
3. In this study, the authors used 10 Drosophila species to test assays. Please mention why they chose them and if they did based on ecological or physiological reasons, please discuss it.
Author Response
Reviewer 1 (response in bold):
We thank reviewer 1 for the useful suggestions given. Responses to individual points are listed below.
- “I know that researches using in Drosophila species (especially melanogaster) have greatly contributed to the understanding of thermal tolerance in insects. However, based on the experimental design and results of this MS, I think it is inappropriate to use that word of “arthropods” in the title. I recommend to change the title.”
As suggested, the title is now changed from “arthropods” to “drosophilids” to better reflect the content of the paper.
- “In Materials and methods parts (L 122-134), please explain the methods to measure time and temperature to enter heat come in more detail. For example, how often did they check the condition of the flies in the bottle? Did they take the bottle out of the water bath for observation?”
The Materials and Methods section has now been expanded to give the reader a better idea of how individual flies were tested.
- “In this study, the authors used 10 Drosophila species to test assays. Please mention why they chose them and if they did based on ecological or physiological reasons, please discuss it.”
We have added more information on the 10 species investigated in the study (please see new supplemental 2). It t now mentioned in the text that species were randomly chosen across the Drosophila phylogeny. We feel that a further discussion of results in relation to ecotype and geographical origin cannot be justified given the low number of species assessed and this is beyond the scope of this study.
Reviewer 2 Report
Dear Authors,
Find the comments as an attached file.

Author Response
Please see attached word document for reply

Reviewer 3 Report
Comments to authors
insects-881127: Comparison of static and dynamic assays when 2 quantifying thermal plasticity of arthropods
GENERAL
The authors address a longstanding question about the comparability of static vs dynamic methods of estimating insect thermal tolerance, under a thermal plasticity scenario. Static and dynamic measured of acute heat stress tolerance was measured for 10 Drosophila species reared under control and two heat hardening treatments. The authors find good correlation between the knockdown time (a static measurement) and critical thermal maximum (a dynamic measurement), and are also able to reasonably predict critical thermal maxima based on measured knockdown times for each species and treatment (despite lack of species-specific z factors for heat injury accumulation). The study provides support for the idea that the two methods measure the same underlying mechanisms, and that they are both valid/relatable approaches (though each may be best suited to a particular scenario).
This paper is very well-written, clear, concise, and overall a pleasure to read. The authors make very good use of the discussion to point out the utility of the findings while exercising reasonable caution. I have only minor typographical/style edits (below).
INTRODUCTION
L46,48: Consider adding a comma after “assay”.
L67: Space missing before “If”.
L69: Remove extra space between “of the”
L72: Perhaps write “method” instead of “methodology” (the latter being the study/analysis of methods).
L87: Slightly awkward wording. Suggest writing as “However, empirical studies do not...”.
METHODS
L116: More conventional to write “...six-day-old adult males...”.
L122 vs 131: Please check over to use consistent notation “6” vs “six”.
L138, 171: Replace x with ×.
L144-145: For less awkward sentence structure, please move “within each species” to after “heat tolerance”.
L148: Likely an extra space in this formula.
L151: Add an apostrophe at the end of species (species’).
L152: A comma before “and” might help with this long sentence.
L154: Remove extra space after “recorded”.
L157: I think there is no need to redefine the acronym.
L164-170: I wonder if the font should be changed back to Times New Roman for this text (or is it supposed to remain in ‘formula font’?).
L167: “repair rate” would be fine, and consider adding a comma after “rate”.
RESULTS
L190: Add a comma after “ramp test”.
L200-201: Consider wording as “Statistically significant hardening effects are underlined and in bold”.
Table 2: Consider switching to the . format for decimal place (as opposed to the , style).
L210: Change to “10 species”.
DISCUSSION
L230: “climate change” is more conventional (compared to ‘changes’).
L232: Consider writing as “...species, employing both static and...”.
L244: Can a treatment be maladaptive? Consider rewording.
L245: Add a comma after “experienced”.
L275: Remove extra space after “suggestion”, and add “the” before “previous” (alternatively write as “in opposition to previous suggestions that...”, if more than one author has suggested).
L277 (and see L284): Write as “...by a ”harden as you go” phenomenon, where...”.
L280: “relationship”.
L297-300: Consider adding commas and/or breaking into two sentences.
L300: “the trade-off with dynamic...”.
Author Response
Reviewer 3:
We thank reviewer 3 for the very useful suggestions made.
Each typographical/style suggestion raised by reviewer 3 have been included into the paper and are highlighted with the “track changes”-function in word of the resubmitted paper.